# pH- and sodium-induced changes in a sodium/proton antiporter

**Cristina Paulino, Werner Kühlbrandt\***

Department of Structural Biology, Max Planck Institute of Biophysics, Frankfurt am Main, Germany

**Abstract** We examined substrate-induced conformational changes in MjNhaP1, an archaeal electroneutral $Na^+/H^+$-antiporter resembling the human antiporter NHE1, by electron crystallography of 2D crystals in a range of physiological pH and $Na^+$ conditions. In the absence of sodium, changes in pH had no major effect. By contrast, changes in $Na^+$ concentration caused a marked conformational change that was largely pH-independent. Crystallographically determined, apparent dissociation constants indicated ~10-fold stronger $Na^+$ binding at pH 8 than at pH 4, consistent with substrate competition for a common ion-binding site. Projection difference maps indicated helix movements by about 2 Å in the 6-helix bundle region of MjNhaP1 that is thought to contain the ion translocation site. We propose that these movements convert the antiporter from the proton-bound, outward-open state to the $Na^+$-bound, inward-open state. Oscillation between the two states would result in rapid $Na^+/H^+$ antiport.

## Introduction

$Na^+/H^+$ antiporters are ubiquitous and essential secondary-active transporters found in the cell membranes of all organisms. They play crucial roles in the regulation of intracellular pH, sodium homeostasis, and cell volume. In humans, $Na^+/H^+$ antiporter dysfunction is associated with numerous serious or life-threatening diseases (*Donowitz et al., 2013*), which makes them important drug targets (*Fliegel, 2009*; *Boedtkjer et al., 2012*; *Loo et al., 2012*). $Na^+/H^+$ antiporters belong to the superfamily of cation/proton antiporters (CPA), which include the CPA1 and CPA2 subfamilies as main branches (*Brett et al., 2005*). CPA1 transporters are electroneutral and most likely exchange protons and $Na^+$ ions with a 1:1 stoichiometry, whereas CPA2 transporters, which mostly exchange two protons per $Na^+$, are electrogenic. The best-known member of the CPA2 subfamily is the $Na^+/H^+$ antiporter NhaA from *E. coli*. EcNhaA enables *E. coli* to survive at high salinity or alkaline pH (*Padan and Schuldiner, 1994*), making use of the proton-motive force to extrude sodium from the cell (*Figure 1*). Other members of the CPA2 subfamily include the plant CHX transporters and the mammalian NHA transporters. Well-known representatives of the CPA1 subfamily include the medically important mammalian NHE exchangers and the archaeal NhaP antiporters (*Brett et al., 2005*).

The activity of $Na^+/H^+$ antiporters is highly dependent on the concentration of their substrate ions, $H^+$ and $Na^+$. For EcNhaA, a maximum transport rate of $10^5$ $min^{-1}$ at pH 8.5 has been reported, which drops by two or three orders of magnitude at pH 6.5 (*Taglicht et al., 1991*; *Rimon et al., 1998*; *Padan, 2008*). This pH dependence has been attributed to a putative cytoplasmic pH sensor, which transfers the antiporter into an acidic-locked conformation at low pH (*Taglicht et al., 1991*; *Padan, 2008*). A recent study of ΔpH and Δ$Na^+$-driven transport indicated a symmetrical bell-shaped pH dependence of EcNhaA (*Mager et al., 2011*). Transport activity was maximal at pH 8.5, and more than doubled as the $Na^+$ concentration increased from 10 mM to 100 mM. These observations are consistent with a simple kinetic model whereby $H^+$ and $Na^+$ compete for the same substrate binding site (*Mager et al., 2011*), thought to include two conserved aspartates in TMH V of EcNhaA. According to this model,

**\*For correspondence:** werner.kuehlbrandt@biophys.mpg.de

**Reviewing editor**: Benoit Roux, University of Chicago, United States

**eLife digest** Antiporters are proteins that move molecules or charged particles, such as sodium ions and protons, in and out of cells. Antiporters therefore have an important role in controlling the conditions inside a cell, such as the pH (which is a measure of acidity) and sodium content (which is a measure of saltiness). In human cells, defects in sodium/proton antiport result in heart or kidney failure and other serious diseases.

Some aspects of sodium/proton antiporters are well understood, such as the levels of saltiness and acidity that trigger the flow of charged particles in and out of bacterial cells, but the details of how sodium ions or protons activate an antiporter are unknown. Now Paulino and Kühlbrandt have used a technique called electron crystallography to study how the structure of the sodium/proton antiporter changes as the acidity or salt conditions sensed by the antiporter vary.

When the concentration of sodium ions was increased at acidic conditions (low pH), the structure of the antiporter began to change so as to increase the ion flow. However, no such changes were observed when the concentration of sodium ions was held constant at a low level while the pH was increased. These findings suggest that, contrary to previous thinking, the operation of a sodium/proton antiporter is largely determined by the concentration of sodium ions. A better understanding of the operation of sodium/proton antiporters should help with the design new treatments for faulty antiporters.

inactivation at low pH occurs because $Na^+$ ions cannot compete effectively against protons for the binding site. Inactivation at high pH is simply due to the depletion of $H^+$ substrate ions at the binding site, as the proton concentration becomes too low to drive transport.

The archaeal CPA1 antiporters share significant sequence homology of functionally important regions with the mammalian NHE antiporters (*Goswami et al., 2011*). A study of NHE1 in mammalian cells has shown a strong dependence of transport activity on both $H^+$ and $Na^+$ concentrations, with activity dropping to background level at pH 8, and a strong dependence on extracellular $Na^+$ (*Fuster et al., 2008*). Like NHE1, but unlike EcNhaA, the $Na^+/H^+$ antiporter NhaP1 from *Methanocaldoccocus jannaschii* (MjNhaP1) is thought to use a sodium gradient to extrude protons from the cell (*Figure 1*) (*Thauer et al., 2008*; *Lee et al., 2012*). Like NHE1 (*Fuster et al., 2008*), but again unlike EcNhaA, MjNhaP1 is active at pH 6, and down-regulated at pH 7.5 or above (*Hellmer et al., 2002*; *Goswami et al., 2011*).

The first insight into the structure of a $Na^+/H^+$ came from the 3D map of the EcNhaA dimer in the membrane at 7 Å resolution, obtained by electron crystallography of 2D crystals (*Williams, 2000*). The map revealed 12 transmembrane α-helices (TMHs) per protomer, referred to as TMH I-XII. The TMHs of EcNhaA were arranged in two groups: a 6-helix bundle at the tip of each protomer, and a row of 6 helices at the dimer interface. The 3.45 Å X-ray structure of monomeric EcNhaA in an inward-open conformation provided further details, including a pair of discontinuous helices (TMH IV and XI) in the 6-helix bundle that were proposed to harbor the ion translocation site (*Hunte et al., 2005*). By electron crystallography we obtained a projection structure (*Vinothkumar et al., 2005*) and, more recently, a 3D map of the MjNhaP1 dimer in the membrane at 7 Å resolution (*Goswami et al., 2011*) that indicated 13 TMHs, referred to as TMHs 1-13. While the dimer interface looked very different to that of EcNhaA, the structure of the 6-helix bundle was similar, supporting the notion of related transport mechanisms.

Two-dimensional (2D) crystals are ideal for investigating conformational changes of membrane proteins in a native-like lipid environment (*Subramaniam et al., 1993*; *Beroukhim and Unwin, 1995*). By this approach we discovered substrate-induced changes in MjNhaP1 (*Vinothkumar et al., 2005*) and in EcNhaA (*Appel et al., 2009*). In both cases, the changes occurred in the 6-helix bundle. Here, we present a detailed study of in situ conformational changes in 2D crystals of MjNhaP1 in response to pH and $Na^+$. To separate the effects of $H^+$ and $Na^+$, 2D crystals were grown without NaCl and examined by electron crystallography in a wide range of carefully controlled pH and ionic conditions. Our results provide new insights into the molecular mechanisms of activation and substrate binding in the CPA1 antiporters.

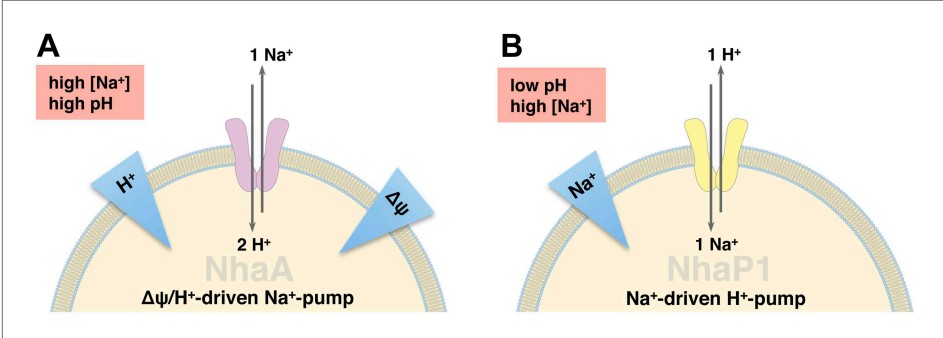

**Figure 1**. Physiological roles of EcNhaA and MjNhaP1. (**A**) EcNhaA is a sodium pump driven by the proton-motive force, exchanging one sodium ion against two protons (*Taglicht et al., 1991*), enabling *E. coli* to survive at high sodium and alkaline pH. (**B**) MjNhaP1 is thought to act mainly as a proton pump driven by the sodium gradient, exchanging one proton against one sodium ion. Like the homologous NHE1 in mammals (*Lee et al., 2012*), MjNhaP1 plays a critical role in pH homeostasis and enables *M. jannaschii* to survive at low pH (4–6) and sodium concentrations up to 0.9 M (*Jones et al., 1983*; *Hellmer et al., 2002*).

## Results

### Electron crystallography and difference maps

For electron crystallography, MjNhaP1 was purified and crystallized under $Na^+$-free conditions. The 2D crystals looked similar to those obtained earlier (*Vinothkumar et al., 2005*; *Goswami et al., 2011*) but were more highly ordered, diffracting up to 6 Å resolution. In total, 29 data sets of MjNhaP1 at different pH and salt concentrations were collected, processed and analyzed (*Supplementary file 1*).

Peaks in the projection maps characteristic of membrane-spanning helices at 6 Å resolution were observed in the two protomers of the crystallographic MjNhaP1 dimer (*Figure 2*). The 7 Å 3D EM model of MjNhaP1 (*Goswami et al., 2011*) enabled us to assign peaks in the projection maps to the TMHs and to the cytoplasmic (c) or exoplasmic (e) halves of the discontinuous TMHs 5 ($5_c/5_e$) and 12 ($12_c/12_e$) in the protomer (*Figure 2*). TMH 1-3 and 7-10 form the dimer interface. Even though tilted or discontinuous TMHs overlap to some extent in projection, the assignment of helices $5_e$, 6 and 13 to well-defined peaks in the projection of the 6-helix bundle was unambiguous. Helix $5_c$ overlaps with helix $12_e$ in one prominent peak in the center of the MjNhaP1 protomer, whereas helix $12_c$ coincides with the projected densities of TMH 4 and 11 (*Figure 2*).

### pH-induced conformational changes

Conformational changes in the MjNhaP1 dimer were visualized by projection difference maps, calculated from amplitudes and phases collected under defined pH and salt conditions. To observe pure pH-induced conformational changes, 2D crystals grown without $Na^+$ were incubated on the EM grid with sodium-free buffers at pH 4, 6 or 8 (*Figure 3*; *Supplementary file 1*). The resulting projection maps indicated only minor differences in the shape or position of density peaks. A small positive peak close to TMH 10 was observed upon a shift from pH 4 or pH 8 to pH 6 (*Figure 3A,B*) and a similarly-sized negative difference peak close to TMH 6 occurred in the transition from pH 8 to pH 6. Differences between pH 4 and pH 8 without $Na^+$ were at background level (*Figure 3C*). These results indicate that a change in pH alone had no major effect on the conformation of MjNhaP1.

### Sodium-induced conformational changes

To examine pure $Na^+$-induced conformational changes, 2D crystals of MjNhaP1 grown without $Na^+$ were incubated with 20–500 mM NaCl at constant pH. Addition of $Na^+$ at pH 8 caused evident changes in the 6-helix bundle (*Figure 4A*), giving rise to major peaks in the difference maps (*Figures 4C and 5A*). The unit cell changed gradually from 81.2 × 104.2 Å to 80.6 × 107.9 Å at 500 mM $Na^+$, whereas a pH shift in the absence of salt had no significant effect on the unit cell dimensions (*Supplementary file 1*).

All difference maps at pH 8 showed the same three sets of positive/negative difference peaks, outlined by ovals in *Figure 5A*. Difference peaks within each set became progressively stronger as the NaCl concentration increased from 20 mM to 500 mM. At 20 mM, there were 7–9 contour levels

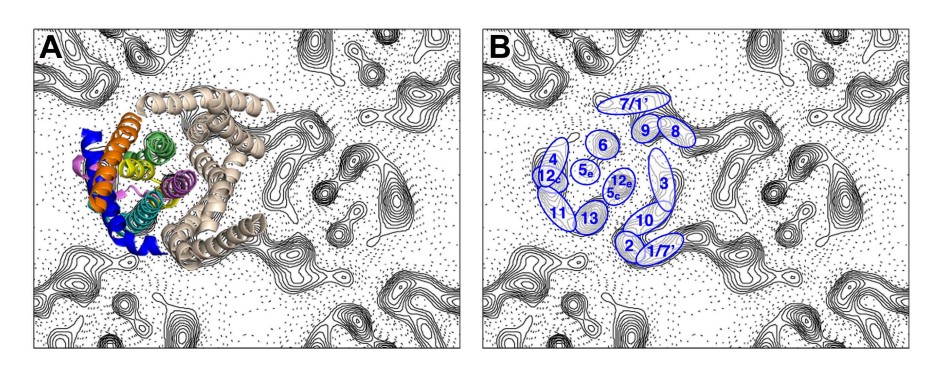

**Figure 2**. Helix assignment. 6 Å projection map of the MjNhaP1 dimer in two-dimensional membrane crystals at pH 4 with 500 mM NaCl. (**A**) Superposition of one protomer of the MjNhaP1 3D EM model (*Goswami et al., 2011*). The model is seen from the extracellular space. The 6-helix bundle is colour-coded, while the 7 TMH at the dimer interface are neutral. (**B**) TMHs are shown as ovals and numbered as in the MjNhaP1 model. Identities of TMH 1 and 7 in the model were ambiguous.

between the lowest negative and highest positive peaks in each set, rising to a peak-to-trough difference of 15 contour levels at 500 mM NaCl (*Figure 5A*). A positive peak that was not matched by a corresponding negative peak appeared at 100 mM NaCl and above (*Figure 5A*), indicating a gain of order in this protein region. Control experiments gave similar results for $Na^+$ and $Li^+$, which are both substrate ions of MjNhaP1, whereas $Mg^{2+}$, which is not transported, did not give rise to significant difference peaks under otherwise similar conditions (*Figure 5—figure supplement 1*). $K^+$, which is likewise not transported but has been reported to bind to EcNhaA (*Alhadeff et al., 2011*), caused changes above background level that were however different from those observed with $Na^+$ or $Li^+$ (*Figure 5—figure supplement 1*). The strong changes observed with $Na^+$ or $Li^+$ are thus substrate-specific.

At pH 4, essentially the same pattern of difference peaks and changes in unit cell dimensions were found as at pH 8 (*Supplementary file 1* and *Figure 5C*). However, these differences occurred at NaCl concentrations that were roughly 10-fold higher than at pH 8 (*Figures 5 and 6*). Difference maps at 20 or 100 mM NaCl at pH 4 were largely featureless, except for one comparatively weak pair (6 or 7 contour levels peak-to-trough) at the position of the strongest peaks at higher NaCl concentration (*Figure 5C*). In addition, at NaCl concentrations of 150 mM or above, there was a broad difference peak to the left of the 6-helix bundle that was not observed at pH 8 (arrow in *Figure 5C*).

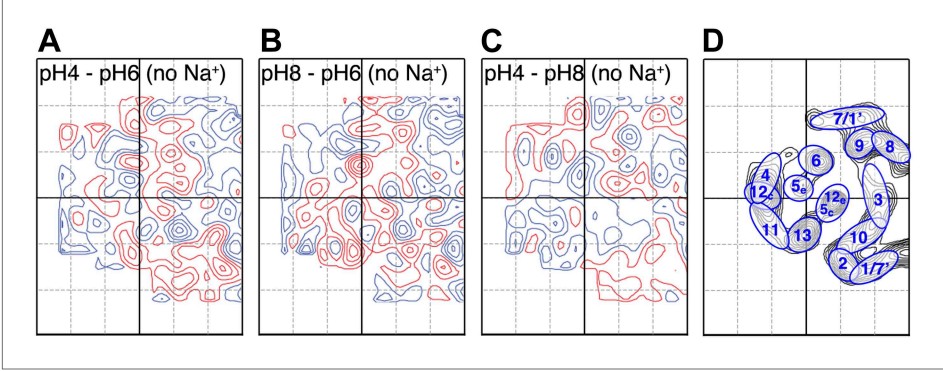

**Figure 3**. pH-induced conformational changes in absence of sodium. Difference maps at 6 Å resolution between projection maps of MjNhaP1 2D crystals grown without sodium at different pH: (**A**) pH 4 minus pH 6; (**B**) pH 8 minus pH 6; and (**C**) pH 4 minus pH 8. (**D**) Helix assignment as in *Figure 2B*. Blue contours indicate positive differences, negative differences are shown in red.

**Figure 4**. Sodium-induced conformational changes at pH 8. (**A**) 6 Å projection maps of MjNhaP1 crystals at pH 8 without sodium (left) and at increasing sodium concentration. (**B**) Helix assignment as in *Figure 2B*. (**C**) Difference map at 6 Å resolution between projection maps of MjNhaP1 at pH 8 in absence of sodium and at 500 mM NaCl. Blue contours indicate positive differences, negative differences are shown in red.

At pH 6, again the same sets of difference peaks were evident, but only at elevated NaCl concentrations of 250 mM or above (*Figure 5B*). Differences were much weaker (maximally 7–8 contour levels peak-to-trough) and more diffuse than at pH 8 or at pH 4 (*Figure 5*), even up to 1 M NaCl (not shown). Unit cell parameters at pH 6 did not change significantly (*Supplementary file 1*).

## Crystallographically derived Na$^+$ dissociation constants

To estimate the sodium affinity of MjNhaP1 under different pH regimes, peak heights for each of the three sets of positive/negative difference pairs shown in *Figure 5* were plotted as a function of Na$^+$ concentration (*Figure 6*). Analysis of these titration curves yielded apparent dissociation constants for Na$^+$ ($K_D^{Na^+,app}$) (*Figure 6C*). At pH 8, where conformational changes were evident already at comparatively low NaCl concentrations, the three sets of difference pairs gave almost identical apparent $K_D^{Na^+}$ values of around 30 mM. At pH 4, the apparent $K_D^{Na^+}$ of ~280 mM indicated roughly 10-fold weaker binding than at pH 8.

## Helix movements

Strong pairs of positive/negative peaks in projection difference maps at 6–8 Å resolution are indicative of helix movements, which can be either lateral displacement or helix tilts (*Subramaniam et al., 1993*; *Appel et al., 2009*). The three sets of strong difference peaks in *Figure 5* all map to the 6-helix bundle of MjNhaP1 and indicate significant and specific rearrangements of TMHs in this part of the protein. The changes take place in the physiological pH and Na$^+$ concentration range for *M. jannaschii* (*Jones et al., 1983*) and thus are likely to reflect the conformational changes in the transport cycle. Our model of MjNhaP1 (*Goswami et al., 2011*) superposed on the projection and difference maps enables us to assign these changes to individual TMHs, in favourable cases even to the side of the membrane on which they occur.

Of the three sets of difference peaks in *Figure 5* that are common to all three pH regimes, the strongest pair (set 1) coincides with the projection of the cytoplasmic half of TMH 5 and the extracellular half of TMH 12 (helix segments 5$_c$/12$_e$), indicating a joint movement of these two half helices. Inspection of the corresponding peaks in the projection map (*Figure 4*) and the 3D model indicates a displacement, most likely a lateral movement, of the 5$_c$/12$_e$ helix pair by ~2 Å towards TMH 10 and 13.

The second-strongest pair of positive/negative difference peaks (set 2 in *Figure 5*) maps to the projection of TMH 6. In *Figure 4*, the peak corresponding to TMH 6 is elongated in the absence of Na$^+$ and becomes round at elevated NaCl concentration, indicating that this helix reorients in response to Na$^+$ binding. In the 3D model the extracellular half of TMH 6 is tilted, so that the entire TMH 6 becomes more straight and perpendicular to the membrane in response to Na$^+$ binding.

The third set of difference peaks in *Figure 5* (set 3) coincides with the map region occupied by TMH 13 and helix 5$_e$. This set consists of a central positive peak, between two smaller negative peaks. In principle, the positive peak could form a pair with either (or both) of the negative peaks, thus indicating a movement of the corresponding helices. Inspection of the projection map (*Figure 4*) helps to resolve this ambiguity. The projection peak corresponding to TMH 13 is more elongated

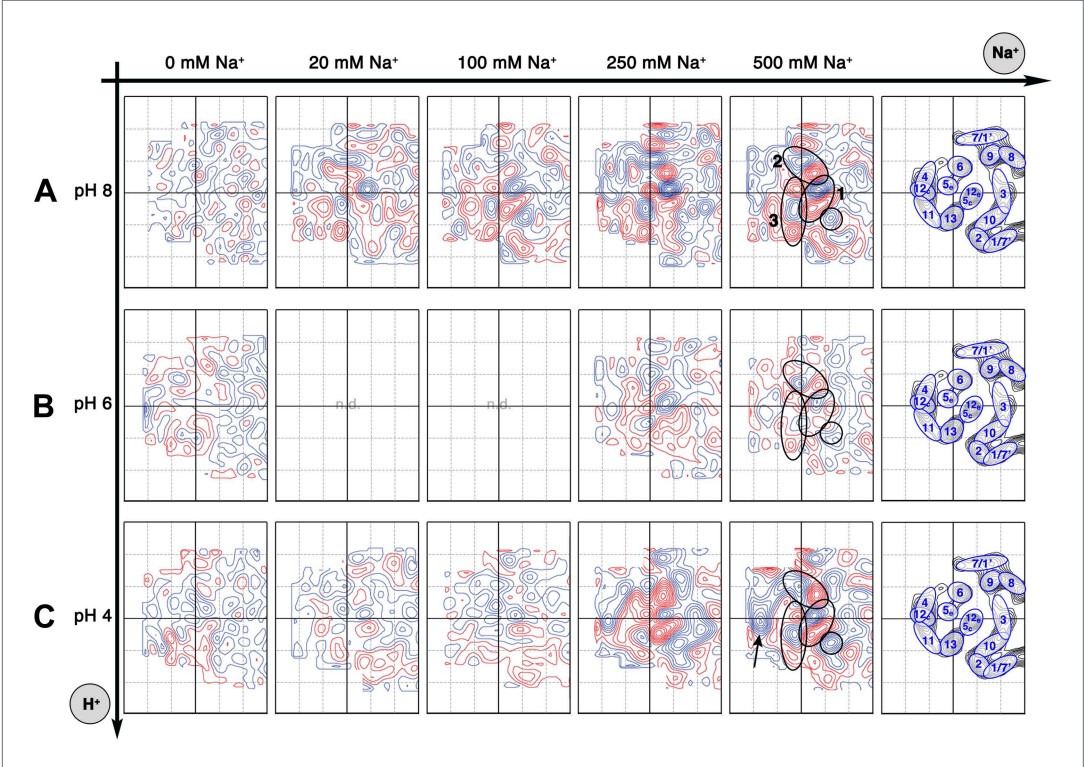

**Figure 5**. Sodium-induced conformational changes at different pH. Matrix of 6 Å difference maps between projections of MjNhaP1 2D crystals in absence of sodium and the sodium concentrations indicated at pH 8 (**A**), pH 6 (**B**) and pH 4 (**C**). Clusters of predominant positive/negative difference peaks are outlined by ovals, single positive difference peaks are circled. The arrow indicates a broad difference peak observed at pH 4. Helices are assigned to regions of the projection maps at the end of each row. The first column shows control difference maps calculated from datasets split randomly into two halves to assess the level of background noise, which was estimated at around ±2 contour lines. Blue contours indicate positive differences, negative differences are shown in red.

The following figure supplements are available for figure 5:

**Figure supplement 1**. Substrate specificity.

**Figure supplement 2**. Difference maps calculated for TtNapA.

**Figure supplement 3**. Comparison to the difference map obtained by *Vinothkumar et al. (2005)*.

at 500 mM NaCl than in absence of sodium, indicating that this helix becomes more tilted with increasing $Na^+$ concentration. A peak for helix $5_e$ is not evident at low $Na^+$, but a clear peak is present at 100 mM NaCl and above in the projection maps. Therefore helix $5_e$ is either highly tilted or disordered at low $Na^+$, which would both make it difficult to see in projection. Upon $Na^+$ binding, it either reorients or becomes ordered and then runs more or less perpendicular to the membrane plane.

The broad difference peak to the left of the 6-helix bundle in *Figure 5C* that appeared at pH 4 above 150 mM NaCl indicates a $Na^+$-induced movement of the group of helices on the outside of the bundle, consisting of TMH 4, 11 and helix $12_c$. Again, inspection of the corresponding projection maps indicates that this part of the bundle shifts by ~2 Å towards the dimer interface. This movement is largely absent at pH 8 or 6. The positive difference peak above 100 mM NaCl at pH 8 or 250 mM at pH 4 (*Figure 5A,C*) indicates a gain of order in the region between the 6-helix bundle and of TMH 10, which is part of the dimer interface. Otherwise, no significant differences were recorded in the central part of the dimer. *Figure 7* summarizes the observed helix movements in the context of the MjNhaP1 model (*Goswami et al., 2011*).

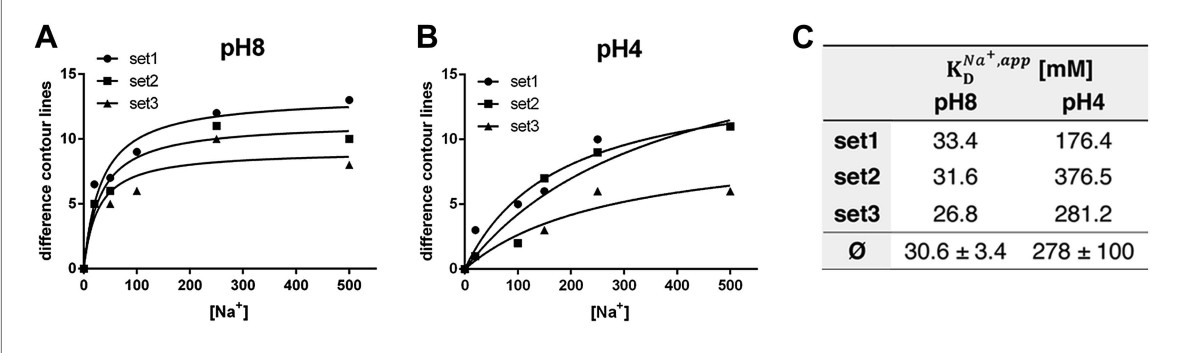

**Figure 6**. pH dependence of apparent dissociation constants for Na+ in MjNhaP1. The number of contour lines (peak-to-trough) was counted for each set of positive/negative difference peaks in the difference maps shown in **Figure 5** including maps generated for 50 mM NaCl at pH 8 and 150 mM NaCl at pH4 (not shown, see **Supplementary file 1**) and plotted against the Na+ concentration at pH 8 (**A**) and pH 4 (**B**). Apparent dissociation constants for Na+ were calculated (**C**). Contour levels observed in the same regions of control difference maps in absence of sodium (first column) were set to zero and subtracted from all subsequent values.

## Discussion

### Comparison with other Na+/H+ antiporters

The *E. coli* antiporter NhaA shares many features with MjNhaP1. Both form dimers of similar shape and size in the membrane, and have the conserved 6-helix bundle thought to harbor the ion translocation site. In EcNhaA and other CPA2 antiporters, the ion-binding and transport site has been proposed to include two conserved aspartates (*Inoue et al., 1995*; *Hunte et al., 2005*; *Maes et al., 2012*). In the CPA1 antiporters, one of the aspartates is replaced by an asparagine, which likewise is thought to participate in ion-binding and translocation (*Hellmer et al., 2003*; *Goswami et al., 2011*). In the EcNhaA X-ray structure (*Hunte et al., 2005*) the two conserved aspartates are located at the center of TMH V, which means that the corresponding residues in MjNhaP1 are in TMH 6. Our difference maps show that this helix participates prominently in Na+-induced changes.

We also found changes in helix $5_e$ of MjNhaP1, which corresponds to the periplasmic half of TMH IV of EcNhaA. Electron crystallography of 2D crystals, side-directed tryptophan fluorescence, ion accessibility studies and MD simulations have all found conformational changes or an intrinsically high degree of flexibility in this half helix (*Appel et al., 2009*; *Kozachkov and Padan, 2011*; *Rimon et al., 2012*), and we find that the corresponding region in MjNhaP1 also moves or rearranges upon Na+ binding. The events known to occur upon ion-binding and translocation in EcNhaA are thus consistent with the conformational changes we observe in MjNhaP1.

Recently, the 3 Å X-ray structure of the Na+/H+ antiporter NapA from *T. thermophilus* has been reported (*Lee et al., 2013*). TtNapA is electrogenic, and thus functionally similar to EcNhaA. However, in terms of protein structure, it resembles MjNhaP1 much more closely, as judged from a 6 Å projection map we calculated from the TtNapA coordinates (*Figure 5—figure supplement 2A* and *Figure 2*). This refers in particular to the position of the 6-helix bundle relative to the dimer interface and the number of TMHs (13 in TtNapA and MjNhaP1, only 12 in EcNhaA). Nevertheless, Lee et al. assume that the structures of TtNapA and EcNhaA show the outward-open and inward-open conformation of essentially the same antiporter. A model of TtNapA in the inward-open conformation based on the EcNhaA structure was generated, and compared it to the outward-open TtNapA structure. Not surprisingly, there were major differences between the inward-open model and the outward-open structure, in particular with respect to the position and orientation of the 6-helix bundle, which appeared to rotate by 20° within the membrane. On the basis of this comparison, they proposed a transport mechanism, which was claimed to apply to all Na+/H+ antiporters. However, such a massive domain movement is difficult to reconcile with the high turnover rates of EcNhaA and the comparatively subtle substrate-induced changes in and around the 6-helix bundle we observe experimentally in MjNhaP1.

We built an inward-open TtNapA model based on the EcNhaA structure, as described by *Lee et al. (2013)*, and calculated a projection difference map between it and the outward-open TtNapA structure

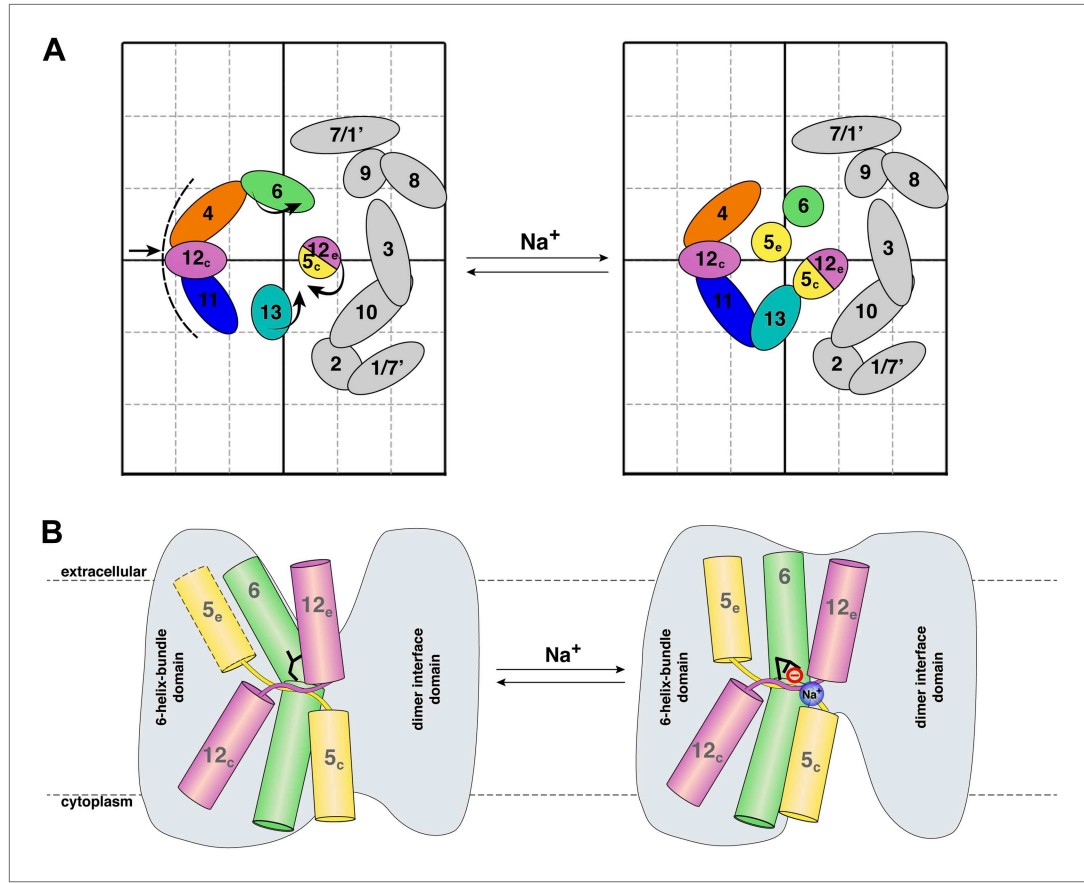

**Figure 7**. Na$^+$-induced conformational changes in MjNhaP1. (**A**) Summary of observed helix movements in response to Na$^+$ binding. Schematic helix positions in absence of NaCl (left) or at pH 4 and pH 8 at >250 mM NaCl (right). Helix movements are indicated by arrows. Helix projections are shown as circles or ovals. The 6-helix bundle is color-coded as in *Figure 2A*. Helices at the dimer interface are grey. (**B**) Model drawing of changes in the positions of TMH 5, 6 and 12 that respond most strongly to Na$^+$ binding. Residue D161 in TMH 6, thought to be directly involved in substrate binding, is shown in black. Na$^+$ binding results in a transition from the apo or proton-bound state, where the putative ion-binding site is likely to be more accessible form the extracellular space (left), to a Na$^+$-bound state, which we propose to be open to the cytoplasm (right).

(*Figure 5—figure supplement 2*). The very large and strong difference peaks in the region of the 6-helix bundle bear no resemblance to *Figures 4 and 5*. We conclude that a 20° in-plane rotation of the 6-helix bundle does not occur in MjNhaP1. It most likely also does not occur in EcNhaA, since *Figure 5—figure supplement 2* also bears no resemblance to projection difference maps obtained under a range of physiological ion and pH conditions with this antiporter (*Appel et al., 2009*).

## pH dependence of sodium binding in MjNhaP1

The projection and difference maps in *Figures 3–5* indicate two different conformations of MjNhaP1: one in the absence of Na$^+$, which looks similar at any pH between 4 and 8, and one in the presence of Na$^+$ above 20 mM at pH 8 or 200 mM at pH 4. The latter shows the Na$^+$-bound state, whereas the former shows the apo or proton-bound state of the antiporter. At elevated Na$^+$ concentrations, the pH 6 maps resemble those at pH 4 or 8, but the difference peaks are less distinct and only roughly half as strong, as would be expected if the corresponding helices were disordered. We ascribe this apparent disorder to the fact that the antiporter is fully active at pH 6, and the helices involved in ion translocation would oscillate between the sodium-free and sodium-bound state. Hence, when the 2D crystals are frozen in liquid nitrogen, the helices are trapped in a continuum of slightly different orientations. In a projection map, which takes the average of all molecules on the 2D crystal lattice, the helices would thus appear disordered, as observed.

At pH 4 and 8 MjNhaP1 is down-regulated (*Vinothkumar et al., 2005*) and presumably more or less rigid. Under these conditions, the effect of Na$^+$ on the structure is progressive, indicating that the maps are averages of an increasing number of molecules on the 2D crystal lattice in the Na$^+$-bound state and a decreasing number in the apo state, as the Na$^+$ concentration increases. The height of the difference peaks is then a direct measure of Na$^+$ affinity, and enables us to determine apparent binding constants for Na$^+$ in different pH regimes (*Figure 6*).

## pH dependence of transport activity

We (*Goswami et al., 2011*) and others (*Hellmer et al., 2002*) have shown that MjNhaP1 is down-regulated at pH 8. Down-regulation at acidic pH has not yet been shown experimentally (largely because there is no suitable pH-sensitive fluorescent dye), but follows from physiological considerations, which indicate that the antiporter must be inactive at pH 4 to prevent uncontrolled influx of Na$^+$ ions that would otherwise result from a large outward pH gradient (*Vinothkumar et al., 2005*). A shutdown at acidic and basic conditions is a key feature of the simple kinetic model proposed by Mager et al. for EcNhaA. This model thus also appears to hold for MjNhaP1. At pH 4, this antiporter may be down-regulated due to competition of H$^+$ and Na$^+$ for a common binding site, and at pH 8 due to proton depletion. Together with the activity peak around pH 6, these two effects would result in a bell-shaped pH dependence of transport activity.

The observed 10-fold increase in apparent $K_D^{Na^+}$ from pH 8 to pH 4 is consistent with a competition of Na$^+$ and H$^+$ for a common binding site, because more Na$^+$ ions are needed to displace the protons as the H$^+$ concentration rises. However, if Na$^+$ can displace H$^+$ and can arrest the antiporter in a Na$^+$-bound state at pH 4, the question arises why this does not also happen at pH 6, where the antiporter is fully active. Apparently, in addition to substrate competition, the transport rate of MjNhaP1 is also modulated in some other way, for example by the protonation states of amino acid sidechains involved in ion-binding and translocation that render the antiporter more flexible at pH 6 than at pH 4 or pH 8. In EcNhaA and NHE1, a pH sensor is thought to transform the antiporters from an active into an acid-locked state at low pH (*Aronson et al., 1982*; *Aronson, 1985*; *Taglicht et al., 1991*; *Wakabayashi et al., 2003*; *Padan et al., 2009*). In the case of EcNhaA, this has been reported to involve a conformational switch at the level of secondary structure (*Herz et al., 2010*; *Diab et al., 2011*; *Schushan et al., 2012*). In MjNhaP1, we see no evidence of a pH-triggered conformational switch (*Figure 3*), so that down-regulation at low pH is not associated with a significant change in secondary structure. However, it may involve the re-orientation of amino acid sidechains, which would not be visible at 6 Å resolution.

In an earlier study, Vinothkumar et al. investigated pH-induced changes in 2D crystals of MjNhaP1 (*Vinothkumar et al., 2005*). The crystals had been grown in the presence of NaCl, and therefore incubating them in salt-free buffers changed both the pH and the Na$^+$ concentration simultaneously. The effect observed by Vinothkumar et al. was thus due to both Na$^+$ and pH, rather than to pH only. We were able to reproduce these differences with 2D crystals grown with or without NaCl, as indicated in *Figure 5—figure supplement 3*.

## Conclusions

By growing 2D crystals of the archaeal Na$^+$/H$^+$ antiporter MjNhaP1 in absence of sodium, we were able to separate effects of the two substrate ions, Na$^+$ and H$^+$, on its conformation. Projection difference maps at 6 Å resolution show that pH in the absence of Na$^+$ has no major effect on the structure of the antiporter. This contrasts with current models of Na$^+$/H$^+$ antiporter regulation, which postulate a significant pH-triggered conformational switch. If such a pH switch occurs in MjNhaP1, it does not affect the secondary structure but may involve sidechain movements, which are not resolved by our method. On the other hand, Na$^+$ ions cause a marked conformational change that is largely pH-independent.

At pH 8 and 4, where MjNhaP1 is down-regulated, the Na$^+$-induced differences reflect a progressive change in the relative population of the two distinct conformational sodium-free and sodium-bound states. This enabled us to deduce apparent binding constants of MjNhaP1 for Na$^+$ under acidic and basic conditions. At pH 6, where this antiporter is fully active, the helices involved in ion translocation appear disordered, due to averaging over many slightly different conformations on the crystal lattice.

The MjNhaP1 model (*Goswami et al., 2011*) allows us to conclude that the helix movements deduced from the projection difference maps in *Figures 4 and 5* change the access to the proposed substrate binding site, as shown schematically in *Figure 7*. In the apo or proton-bound state, the ion-binding site near the center of TMH 6 would be accessible from the extracellular side, making this the

outward-open state. The helix movements brought about by Na$^+$ ions then convert the antiporter into the Na$^+$-bound, inward-open state. Oscillation between the two states would result in Na$^+$/H$^+$ antiport, as specified by the alternating access mechanism (*Jardetzky, 1966*). It is likely that the mammalian NHE antiporters, which share many similarities with the archaeal Na$^+$/H$^+$ exchangers, work in the same way.

## Materials and methods

### Expression and purification

MjNhaP1-His was expressed in the pET26b vector and purified by Ni-NTA affinity chromatography as described previously (*Goswami et al., 2011*), with the following modifications: The Ni-NTA column was washed with 10 cv (column volumes) of buffer 1 (15 mM Tris/HCl pH 7.5, 500 mM NaCl, 15 mM imidazole and 0.03% DDM), followed by 8 cv of sodium-free buffer 2 (15 mM Tris/HCl pH 7.5, 200 mM KCl and 0.03% DDM). MjNhaP1 was eluted with 50 mM K$^+$ acetate pH 4, 100 mM KCl, 5 mM MgCl$_2$ and 0.03% DDM, concentrated and stored at −80°C.

### 2D crystallization and in situ conformational changes

Two-dimensional crystallization of MjNhaP1 with *E. coli* polar lipids (Avanti Polar lipids) was carried out at a final protein concentration of 1 mg/ml, and a final DM concentration of 0.15%. The mixture was incubated for 1.5 h at room temperature and transferred to a dialysis bag with a 14 kDa cutoff. Dialysis was performed at 37°C over 7–10 days, in 25 mM K$^+$ acetate pH 4, 200 mM KCl, 5% glycerol and 5% 2-4-methylpentanediol (MPD). Crystals were obtained at a lipid-to-protein (LPR) range of 0.3–0.7 and were stable for several months. Samples were prepared by the back-injection method (*Wang and Kuhlbrandt, 1991*) in 4% trehalose. The composition of the embedding medium used varied, accordingly. In total, a combination of three different buffers (25 mM K$^+$ acetate pH 4, 50 mM MES pH 6 and 50 mM Tris/HCl pH 8) and four different salts (NaCl, LiCl, KCl, and MgCl$_2$) at different salt concentrations were used (*Supplementary file 1*). 1.5 μl sample was applied to the carbon film, mixed thoroughly with excess of embedding buffer and incubated for 1 min. The grids were blotted and rapidly frozen in liquid nitrogen.

### Data collection and image processing

Images were recorded on Kodak SO-163 film with a JEOL 3000 SFF electron microscope at a nominal temperature of 4 K, an acceleration voltage of 300 kV, a magnification of x 53,000 in spot scan mode at 0.2–1 μm defocus. Crystal quality was evaluated by optical diffraction. Well-ordered areas of 4000 × 4000 or 6000 × 6000 pixels were digitized at 7 μm step size on a Zeiss SCAI scanner. Images were processed using the 2dx software (*Gipson et al., 2007*) and data quality was improved by synthetic unbending (*Arheit et al., 2012*). Projection maps were calculated from at least six lattices and were of similar quality to 6 Å resolution (*Figure 4*, *Supplementary file 1*).

### Difference maps

For calculation of difference maps scripts from the CCP4 program suite package were manually modified and used at each step as indicated in brackets. Projection phases and amplitudes for each data set were calculated (2dx software), scaled (sftools) and subtracted (overlapmap). Difference maps were plotted to reveal pH- or salt-induced conformational changes. To account for small differences in unit cell length, maps were excised (fft, mapmask, maprot, and npo) and placed into the same cell before subtraction. For an estimate of background noise level, image data collected under identical conditions were randomly divided into two sets (*Figure 5* first column), or image data from different conditions were mixed and randomly merged into two separate data sets. These control difference maps were plotted with a stepsize of 1.5 σ between contour levels, indicating a background level of ±2 contour levels, or ~3 σ. The background level was set to be the same in all calculated difference maps by applying the same absolute plotting parameters as used for the control difference maps (*Figures 3–5*).

## Acknowledgements

The authors thank Deryck Mills for maintenance and management of the EM facility, Özkan Yildiz for computational assistance, and Klaus Fendler for helpful discussions. Henning Stahlberg and Marcel Arheit (Biozentrum Basel) provided generous software support.

## Additional information

### Competing interests

WK: Reviewing editor, *eLife*. The other author declares that no competing interests exist.

### Funding

| Funder | Author |
|---|---|
| Max Planck Society | Werner Kühlbrandt |
| Cluster of Excellence–Macromolecular Complexes, Frankfurt | Werner Kühlbrandt |
| FCT PhD fellowship (Fundação para a Ciência e a Tecnologia, Portugal) | Cristina Paulino |

The funders had no role in study design, data collection and interpretation, or the decision to submit the work for publication.

### Author contributions

CP, Conception and design, Acquisition of data, Analysis and interpretation of data, Drafting or revising the article; WK, Conception and design, Drafting or revising the article

## Additional files

### Supplementary files

• Supplementary file 1. Electron crystallographic data. Lower table legend: [a] Calculated from the program AVRGAMPS: mean weighted squared distance of the phase values from the averaged value (90° = random) [b] Calculated from the program FOMSTATS: averaged phase values from the symmetry-constrained target values of 0° or 180° (45° = random). [a,b] Reflections with IQ ≤ 7 Å were included.

### Major dataset

The following previously published dataset was used:

| Author(s) | Year | Dataset title | Dataset ID and/or URL | Database, license, and accessibility information |
|---|---|---|---|---|
| Lee C, Kang HJ, Von Ballmoos C, Newstead S, Uzdavinys P, Dotson DL, Iwata S, Beckstein O, Cameron AD, Drew D | 2013 | Crystal structure of the sodium proton antiporter, NapA | 4BWZ; http://www.pdb.org/pdb/explore/explore.do?structureId=4BWZ | Publicly available at RCSB Protein Data Bank (http://www.rcsb.org). |

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
