## [Decision Letter]

Thank you for sending your work entitled “pH- and sodium-induced changes in a sodium/proton antiporter” for consideration at *eLife*. Your article has been favorably evaluated by a Senior editor and 3 reviewers, one of whom is a member of our Board of Reviewing Editors. We believe that the manuscript should be revised to address the comments below. The Reviewing editor and the other reviewers discussed their comments before we reached this decision, and the Reviewing editor has assembled the following comments to help you prepare a revised submission.

1) A striking result is the inconsistency with the concept of allosteric pH regulator. The pH dependence of Na^+^/H^+^-antiporters has been attributed to an allosteric pH regulator (pH sensor) believed to transform the antiporter from an active into a acid-locked state at low pH. Because no major conformational changes are observed here upon pH shift in the absence of Na^+^, the current results do not support this model for MjNhaP1. Instead, the observed changes appear to be a consequence of protons and substrate ions competing for the same binding site. The authors state that this minimal kinetic model proposed for NhaA by [27] accounts for most of their observations, although the pH dependence of the Na^+^ effects on the structure of MjNhaP1 is clearly not linear. If they were, changes at pH 6 would be expected to be somewhere in between those at pH 4 and pH 8, which is evidently not the case. It would be nice to detail this point a little more quantitatively. If one assumes that Na^+^ can only bind when the carboxylate groups are deprotonated, then it seems that the apparent binding constant of Na^+^ to the proton would depend (in a non-linear fashion) on the pH. Could the authors expand along those lines?

2) The apparent K_D_ at pH 8 is around 30 mM, around 280 mM at pH 4. How does that compare with functional measurements of a transporter in a fluid membrane (as opposed to a 2D crystal assembly)? This also raises an interesting question concerning the conformational states underlying the present data. A conformational change is observed as a function of Na^+^ concentration. The 2D projection thus represents an ensemble average over a large ensemble of molecules. Experimentally, the conformational changes that are observed as a progressive effect of Na^+^ could, therefore, reflect either a progressive change in the actual protein conformation itself, or a change in relative population of the two distinct conformational sodium-bound and sodium-unbound states. The latter seems reasonable, but it would be nice to make this point clearer on the basis of the present analysis.

3) While the results seem convincing, one is left unsure about the interpretation of the results obtained at pH 6. First, the authors conclude that sodium ions bind only very weakly (K_D_ estimated at 500 mM) to the transporter at pH 6. Interestingly, even in the absence of sodium ions, the structures of the transporter are more similar at pH 4 and 8 than at pH 6. It would be essential for the authors to propose a specific model that could account for a bell-shaped pH dependence of the transporter properties, including affinity for sodium and, perhaps, structure. It is unclear what could be the origin of such dependence. Furthermore, it seems impossible, unless more than one proton binds to the protein.

4) The authors state that the lack of large pH-dependent structural changes is inconsistent with allosteric regulation by protons. However, such regulation may not require large structural changes. There is confusion about the concept of the “substrate inhibited state”. It seems that sodium-inhibited state may only occur if the cytoplasmic sodium concentration were high (above 300 mM at pH 4 or 30 mM at pH 8). Similarly, proton-inhibited state would only occur if the outside pH were low. In contrast, high extracellular Na and low cytoplasmic pH should not lead to inhibition. Collectively, the mechanistic origin and the physiological need for the bell-shaped pH dependence of sodium affinity and structure remained unclear.

5) The authors cite a recent paper that states that the reduction in activity in NhaA due to pH is simple competition between protons and Na^+^. There are issues with the implications for the results on NhaA. Some may disagree with this interpretation of the paper due to two factors: how can a change in three orders of magnitude in activity be observed when the concentration of a competing agent is changed by 1.5 pH units? Why do we find very close homologues of NhaA (50% and higher) that work nicely in acidic conditions? Can we assume that they have developed an entirely new mechanism?

---

## [Author Response]

*1) A striking result is the inconsistency with the concept of allosteric pH regulator. The pH dependence of Na*^*+*^*/H*^*+*^*-antiporters has been attributed to an allosteric pH regulator (pH sensor) believed to transform the antiporter from an active into a acid-locked state at low pH. Because no major conformational changes are observed here upon pH shift in the absence of Na*^*+*^*, the current results do not support this model for MjNhaP1. Instead, the observed changes appear to be a consequence of protons and substrate ions competing for the same binding site. The authors state that this minimal kinetic model proposed for NhaA by*
[27]
*accounts for most of their observations, although the pH dependence of the Na*^*+*^
*effects on the structure of MjNhaP1 is clearly not linear. If they were, changes at pH 6 would be expected to be somewhere in between those at pH 4 and pH 8, which is evidently not the case. It would be nice to detail this point a little more quantitatively. If one assumes that Na*^*+*^
*can only bind when the carboxylate groups are deprotonated, then it seems that the apparent binding constant of Na*^*+*^
*to the proton would depend (in a non-linear fashion) on the pH. Could the authors expand along those lines*?

The projection maps indicate two different conformational states of the antiporter: one in the absence of Na^+^, which looks similar at any pH between 4 and 8, and represents the apo or proton-bound state. The Na^+^-bound state is obtained at elevated Na^+^ concentrations, and looks similar at pH4 and 8. At pH 4 and 8 MjNhaP1 is down-regulated (37) and therefore presumably more or less rigid. The projection maps are thus averages of an increasing number of molecules on the 2D crystal lattice in the Na^+^-bound state and a decreasing number in the apo state, as the Na^+^ concentration increases. The height of the difference peaks is then a direct measure of Na^+^ affinity, and enables us to determine apparent binding constants for Na^+^ in different pH regimes (Figure 6).

The projection difference maps at pH 6 at elevated Na^+^ concentrations are essentially also similar to those at pH 4 and 8, but the difference peaks are weaker and less distinct, suggesting that the helices are disordered. We ascribe this apparent disorder to the fact that at pH 6, where the antiporter is maximally active, the helices involved in ion translocation move rapidly and are trapped in a continuum of slightly different orientations when the 2D crystals are frozen in liquid nitrogen. In a projection map, which takes the average of all molecules on the 2D crystal lattice, the helices would then appear disordered, as observed. This is now discussed in the section ‘pH dependence of sodium binding in MjNhaP1’ of the revised manuscript. Under these conditions, the height of difference peaks is not a direct measure of Na^+^ affinity, so that we cannot deduce an apparent binding constant from the difference peaks at pH 6. Accordingly, we have omitted the estimate of the binding constant at pH 6 from Figure 6.

Since we cannot measure the apparent K_D_ for Na^+^ at pH 6, we can also not compare it to the apparent K_D_^Na^ obtained at pH4 and pH 8. We therefore cannot say at this stage whether the pH dependence of the Na^+^ binding affinity of MjNhaP1 is linear or non-linear (for example, bell-shaped).

By contrast, the pH dependence of the MjNhaP1 transport activity is indeed bell-shaped. We (14) and others (15) have shown that this antiporter is essentially inactive at pH 8. Down-regulation at acidic pH has not yet been shown experimentally (largely because there is no suitable pH-sensitive fluorescent dye), but follows from considerations of cell physiology, which indicate that the antiporter has to shut down at low pH (see [37]). A shutdown at acidic and basic conditions is also a key feature of the simple kinetic model proposed by [27] for NhaA that appears to hold equally well for MjNhaP1. By this model the antiporter is down-regulated at pH 4 due to competition of H^+^ and Na^+^ for a common binding site, and at pH 8 due to proton depletion. Together, these two effects result in a bell-shaped pH dependence of transport activity.

The kinetic model of [27] also accounts for the observed 10-fold increase of the apparent K_D_ for Na^+^ from pH 8 to pH 4, which means that more Na^+^ ions are needed to displace the protons from the common ion-binding site at increasing H^+^ concentration. However, if Na^+^ can displace H^+^ and arrest the antiporter in a Na^+^-bound state at pH 4, the question arises why this does not also happen at pH 6, where the antiporter is fully active. The simplest explanation is that, in addition to substrate competition, the transport rate of MjNhaP1 is also modulated in some other way, for example by the protonation states of amino acid sidechains involved in ion binding and translocation that render the antiporter more flexible at pH 6 than at pH 4 or pH 8 (see also response to comment 4 below). This is now explained more clearly in the section ‘pH dependence of transport activity’ of the revised manuscript.

Any more detailed understanding of the mechanistic origin of the pH dependence in MjNhaP1 and related antiporters must await further kinetic and structural analysis. In particular a high-resolution structure of a Na^+^/H^+^ antiporter that resolves the bound substrate ions and the residues involved in substrate binding and translocation is needed.

*2) The apparent K*_*D*_
*at pH 8 is around 30 mM, around 280 mM at pH 4. How does that compare with functional measurements of a transporter in a fluid membrane (as opposed to a 2D crystal assembly)*?

As far as we are aware, there are no published kinetic data on MjNahP1. However, in collaboration with the group of Klaus Fendler at our institute we are conducting such measurements with solid-supported membranes that will be reported elsewhere. Initial results indicate a K_D_^Na+^ of 14 mM at high pH (extrapolated to an effective absence of protons). Considering the very different methods by which the binding constants were determined, this is in excellent agreement with the apparent K_D_^Na+^ of 30 mM at pH 8 reported here.

*This also raises an interesting question concerning the conformational states underlying the present data. A conformational change is observed as a function of Na*^*+*^
*concentration. The 2D projection thus represents an ensemble average over a large ensemble of molecules. Experimentally, the conformational changes that are observed as a progressive effect of Na*^*+*^
*could, therefore, reflect either a progressive change in the actual protein conformation itself, or a change in relative population of the two distinct conformational sodium-bound and sodium-unbound states. The latter seems reasonable, but it would be nice to make this point clearer on the basis of the present analysis*.

This is now explained more clearly in our response to comment 1 above, and in the section ‘pH dependence of transport activity’ and Conclusions section of the revised manuscript.

*3) While the results seem convincing, one is left unsure about the interpretation of the results obtained at pH 6. First, the authors conclude that sodium ions bind only very weakly (K*_*D*_
*estimated at 500 mM) to the transporter at pH 6. Interestingly, even in the absence of sodium ions, the structures of the transporter are more similar at pH 4 and 8 than at pH 6. It would be essential for the authors to propose a specific model that could account for a bell-shaped pH dependence of the transporter properties, including affinity for sodium and, perhaps, structure. It is unclear what could be the origin of such dependence. Furthermore, it seems impossible, unless more than one proton binds to the protein*.

See response to comment 1 above.

*4) The authors state that the lack of large pH-dependent structural changes is inconsistent with allosteric regulation by protons. However, such regulation may not require large structural changes*.

An allosteric pH regulator, also referred to as a pH sensor or H^+^-modifier, has been proposed for NhaA and NHE1. This pH sensor is thought to transform the antiporters from an active into an acid-locked state at low pH (5; 4; 35; 38; 29). In the case of NhaA this has been reported to involve conformational changes at the level of secondary structure (17; 9; 33). In MjNhaP1, we found no evidence of pH-induced secondary structure changes in the absence of Na^+^. Instead, we find that Na^+^ binding induces significant changes in helix arrangements in the 6-helix bundle. This does not rule out pH regulation by side chain rearrangements, as explained in the section ‘pH dependence of transport activity’ and Conclusions section of the revised manuscript.

*There is confusion about the concept of the “substrate inhibited state”. It seems that sodium-inhibited state may only occur if the cytoplasmic sodium concentration were high (above 300 mM at pH 4 or 30 mM at pH 8). Similarly, proton-inhibited state would only occur if the outside pH were low. In contrast, high extracellular Na and low cytoplasmic pH should not lead to inhibition. Collectively, the mechanistic origin and the physiological need for the bell-shaped pH dependence of sodium affinity and structure remained unclear*.

Since the orientation of the MjNhaP1 dimers in the 2D crystals is symmetrical (equal numbers of dimers pointing up and down in the crystalline membrane), we cannot say whether the substrate-inhibited state we observe crystallographically occurs in response to an increase in cytoplasmic or extracellular Na^+^ concentration. See also response to comment 1 above.

*5) The authors cite a recent paper that states that the reduction in activity in NhaA due to pH is simple competition between protons and Na*^*+*^*. There are issues with the implications for the results on NhaA. Some may disagree with this interpretation of the paper due to two factors: how can a change in three orders of magnitude in activity be observed when the concentration of a competing agent is changed by 1.5 pH units*?

Please note that MjNhaP1 is not NhaA. First of all, NhaA is electrogenic, with a transport stoichiometry of 2 H^+^ per 1 Na^+^, whereas MjNhaP1 is not. Thus, in the case of NhaA, at least two protonation events have to be considered, which are likely to both affect transport activity, perhaps in different ways.

Second, no one, ourselves included, knows by how much the transport activity of MjNhaP1 changes in response to pH, as this requires precise measurements of the transport rate, which have not been reported. Indeed, in the absence of other factors (such as allosteric regulation), a change by one pH unit can result in a change of activity by maximally one order of magnitude, since the transport activity can maximally increase by a factor of 10 if the substrate concentration rises 10-fold.

Note that the increase in NhaA transport activity by about three orders of magnitude reported by [35] occurred in response to a change in 2, not 1.5 pH units (1.2 umol/min/mg at pH 6.5 to 2210 umol/min/mg at pH 8.5). Interestingly, similar measurements for NhaA reported 7 years later by the same group using the same methodology (31) found an activity increase of only about 2 orders of magnitude in response to a pH change of 2 units (10.2 nmol/mg/min at pH 6.5 to 1700 nmol/mg/min at pH 8.5). By a different experimental approach, [27] found that a change in one pH unit increases the NhaA transport activity by about one order of magnitude, in line with their kinetic model. The change in activity of EcNhaA by three orders of magnitude in response to a pH shift by 2 units may thus be an overestimate.

As for MjNhaP1, no published data on the pH dependence of its transport rate are available at all. Experiments to determine this dependence are currently under way in our laboratory, but they go well beyond the scope of our present electron crystallographic study and will be reported elsewhere.

*Why do we find very close homologues of NhaA (50% and higher) that work nicely in acidic conditions? Can we assume that they have developed an entirely new mechanism*?

We are not sure which NhaA homologues “that work nicely in acidic conditions” the referees are thinking of. [27] showed that NhaA remains active at acidic conditions down to pH 5. They found that competition of H^+^ and Na^+^ for the same binding site only occurs at lower pH at the Na^+^ uptake side, whereas a decrease in pH on the opposite, Na-release side of the antiporter does not inhibit transport, as explained by their kinetic model. A shift in the pH profile can result simply from changes in binding affinity (for Na^+^ or H^+^). This is demonstrated by the NhaA mutant G338S, which has an activity profile shifted by 1.8 pH units towards the acidic range (27), and evidently does not require an entirely new mechanism.